# The Application of the Inbound Marketing Strategy on Costa del Sol Planning & Tourism Board. Lessons for Post-COVID-19 Revival

**Eva M. Sánchez-Teba** [1] [ID], **Josefa García-Mestanza** [2] [ID] and **Mercedes Rodríguez-Fernández** [3,*] [ID]

1   Department of Economics and Business Administration, Faculty of Economics and Business, Campus El Ejido s/n 29071, University of Málaga, 29017 Malaga, Spain; emsancheteba@uma.es
2   Department of Economics and Business Administration, Faculty of Tourism, Campus Teatinos, University of Málaga, 29017 Malaga, Spain; jgm@uma.es
3   Department of Economics and Business Administration, Faculty of Social and Labour Studies, Ampliación Campus Teatinos, University of Málaga, 29017 Malaga, Spain
*   Correspondence: mmrodriguez@uma.es

**Abstract:** The digital era has radically changed the context in which the tourist service is delivered and experienced, changing the decision processes of consumer and company business models. It is necessary to contact the tourism services customer with non-intrusive techniques, at the beginning of their purchase process and accompany them until the final transaction. The main objective of this case study is to analyze the methodology of inbound marketing and show how The Costa del Sol Planning & Tourism Board could work on its sustainable customer relationship model under the concept of seducing the tourist by being pioneers in the application of this strategy to attract tourism after the pandemic caused by COVID-19. Conclusions of this study include measures to restore travelers' confidence which will play an important role in attracting tourists after crisis. An inbound marketing strategy will provide a response as it is based on contact with the future tourist through highly specialized content.

**Keywords:** sustainable strategy of inbound marketing; customer focus; customer experience; digital marketing; smart destination; case study; COVID-19; Costa del Sol

## 1. Introduction

The arrival of technology as a transversal axis in all sectors of the economy obliges the business world to adapt its management techniques, accepting the digital transformation which affects the organization and its industry structure [1]. It also presents a challenge to be flexible in response to market demands. According to [2], the ascendancy of new technologies becomes manifest in the early eighties with the start of global distribution systems and central reserves. However, the beginning of the revolutionary change experienced by the tourism industry, modifying its marketing of tourism products and services [3–5], really began when the internet went global.

Currently tourists are moving in a new context [6] which has transformed tourism-brokering and changed the way tourists consume [7]. Today's technologically informed client tends to look for experiences and has skills using devices which enable them to obtain information through many different channels [8,9]. We now have travellers who use technology to find an e-destination, or interact with it and create their own experiences [6,10,11].

In other words, coming into play in this highly technological context is the customization of products and services [12,13]. Tourists have, thanks to technological tools, been empowered [14], and seek to create their own experiences both online and offline [13] at all stages of their journey [13,15].

Needless to say, the tourism sector must implement digital marketing as one of their strategies. The new paradigm requires methodologies to change the way we communicate, reinforcing the messages that reach us every day (traditional marketing or outbound marketing), and which are sometimes of little use, with others that attract consumers, creating a win/win connection between the brand and the client (inbound marketing).

Marketing faces a crucial moment, in which new theoretical approaches try to redefine its activities, steering it towards methodologies that, beyond commercial aspects, are concerned with establishing new mechanisms of communication with complex human beings [16].

In this context, the emergence of COVID-19 on the world stage has generated what [17] have called *Extended Reality* in their research letter by examining how evolving market-driven factors arising from this magnitude of change will reshape tourism [18]. Authors such as [19] have researched the policy focus for tourism recovery after the outbreak of COVID-19. Along the same lines, our work aims to help provide ideas to overcome the crisis of tourism after the pandemic, contributing notions that help intermediaries, related companies and interest groups, such as tourism employees, local communities, tourism entrepreneurs or tourism educators so that they can withstand the crisis situation until the time of recovery [20].

The aim of this article is to present the data collected at the Costa del Sol Planning and Tourism Board as part of the Inbound Marketing strategy followed by its managers. Likewise, we contribute ideas so that these data can serve as an element of attraction for new tourists to overcome the severe crisis caused by the pandemic.

We consider that this work covers an important gap in the academic literature on the subject of Inbound-Outbound Marketing, as it was not previously studied from the practical perspective that we propose here. Being a case report, this work is a pioneer in offering real empirical data on the sector for the period 2Q 2017-1Q 2019. Not until now had the strategy of Inbound Marketing been applied to Costa del Sol. Furthermore, the situation of the pandemic has produced a cut in the industry and, consequently, in the collection of data. For this reason we present this data as a basis for rebuilding the sector and providing ideas to attract tourists after the COVID-19 crisis.

The following pages are structured as follows: firstly, we give a definition of inbound marketing, differentiating it from outbound marketing. To continue with the methodology used, the case study applied to the Planning and Tourism Board of the Costa del Sol. Finally, results, discussion, conclusions and future research are presented.

## 2. Literature Review

### 2.1. Inbound Marketing. Definition

The origin of inbound marketing can be found in the enormous change that technology has brought about in the purchasing process.

Before the appearance of the internet, most information users had come from companies, while additional information was sought by, for example, asking others about a product or service, or through specialized magazines.

From the 90s onwards, the process for finding information began to change radically. The internet meant much more than an open space for a wider public—it spread to B2B (Business to Business) companies and ecommerce. Search engines and forums appeared, along with blogs in which products, brands and prices were compared, as well as social networks, where users share information about a particular product or service [21]. This led to full transparency and made subjective aspects, like users' opinions, take on great importance in the purchasing process.

Figure 1 shows the purchasing process, or the customer journey, an important term in the development of an inbound marketing strategy. The phases of this model remain the same, but habits in the purchasing process do not. Nowadays, the first three phases (awareness, investigation and decision) are done over the internet, while in the last stage (action), ecommerce is being increasingly used.

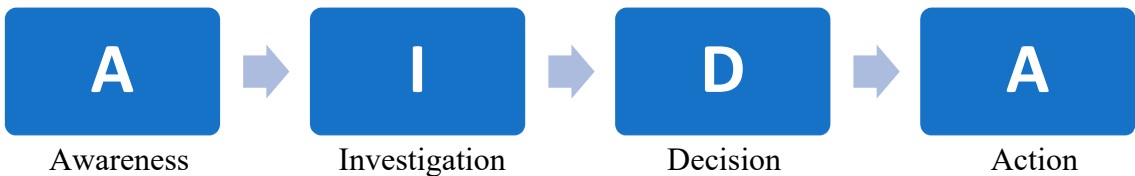

**Figure 1.** Purchasing process. Source: Inbound Cycle, 2018.

The market has had no choice but to adapt to the changes in the purchasing process. Inbound marketing is the response to changes in purchasing process habits. It represents a new sales process that, through technology, gets useful information to the user during the purchasing process, at the moment they need it [21].

This term was coined in 2009 by Brian Halligan, owner of the HubSpot company. His approach is based on the development of a series of related actions which aimed to attract users by providing useful information during the entire purchasing cycle [22]. The aim is to exert influence on the buyer from the moment they have a concrete need until a purchase is made, and accompany them throughout the process.

According to [23] the nature of inbound marketing is to change the role of advertiser to that of socializer, and that a prospective client's attention is caught by content meaningful to them and not by indiscriminate adverts. Even more so, this role of socializer is extremely important in attracting tourists after the Coronavirus pandemic, since now it is necessary to create confidence in the destination directly at the country of origin. Furthermore, after the crisis, an advertiser or socializer could be better considered an advisor that generates confidence in the potential tourist.

*2.2. Inbound Marketing vs. Outbound Marketing*

There is no agreement among the authors who have written about inbound marketing, regarding whether this strategy is mutually exclusive with mass media advertising techniques (outbound marketing) or, on the contrary, it is in fact complementary [24]. According to [22], outbound marketing belongs to the past and inbound marketing will be the predominant strategy in the future. On the other hand, [23] believe that the latter does not replace conventional advertising, but complements it from the digital world. However, there are authors who point out that companies with large resources should not ignore the benefits of reaching a broader audience through outbound marketing [25].

According to [24], inbound marketing tries to create useful content that is remarkable. This type of marketing has a multichannel approach to sharing content and focuses on developing this through blogs and interaction on social networks, among other ways [26,27]. Thus, outbound marketing methodology focuses on high conversion rates and a widespread brand exposure through ads in print and other types of media [28]. In our opinion, outbound marketing can help generate trust among potential visitors, since all its traditional methodology would bring the destination closer to the potential visitor, thereby achieving the higher level of trust needed after the pandemic.

In any case, both methodologies do not have to be opposed to one another. Each contributes something in marketing strategy, although all authors agree on the low cost of inbound marketing. With inbound marketing, the cost per lead is reduced by 62% and with it the cost of acquiring clients [24]. The greatest effort must be made in the creation of contents according to the needs of each client. This involves additional creative effort, as well as investigation and the all-important follow-up, and, while some companies have staff to handle these tasks, others are forced to outsource them.

Another determinant characteristic of inbound marketing is the information provided by the measurement of online actions, something that is not always possible in the offline world. This means that there is much more response capacity and the ability to make decisions in real time and make the necessary changes when the strategy and contents are not giving the desired results [23]. The flexible

character offered by inbound marketing is perfectly adaptable to the needs of today's sustainable tourism to overcome the crisis of COVID-19 (Table 1).

**Table 1.** Inbound marketing vs. Outbound marketing.

| Inbound Marketing (IM) | Outbound Marketing (OM) |
| --- | --- |
| | Strategy mutually exclusive with mass media advertising techniques |
| Complementary strategies | |
| Future | Past |
| IM does not replace conventional advertising, but complements it from the digital world<br><br>IM tries to create useful content that is remarkable | Companies with large resources should not ignore the benefits of reaching a broader audience through OM<br>OM methodology focuses on high conversion rates and a widespread brand exposure through ads in print and other types of media<br>OM can help generate trust among potential visitors, since all its traditional methodology would bring the destination closer to the potential visitor thereby achieving the higher level of trust needed after the pandemic |
| Both methodologies do not have to be opposed to one another | |
| Low cost of IM<br>Characteristic of IM is the information provided by the measurement of online actions, something that is not always possible in the offline world<br>Much more response capacity<br>Ability to make decisions in real time<br>The flexible character offered by IM is perfectly adaptable to the needs of today's sustainable tourism to overcome the crisis of COVID-19 | |

Source: Own creation.

### 2.3. Inbound Marketing: Process

Besides providing support through its user-centered philosophy and value to all users, inbound marketing is seen as an effective methodology adapted to current consumer habits [29].

As mentioned, only by providing useful user content will we be able to develop the purchasing process, in its different phases, starting from the first visit to a company's website, until, if necessary, the close of sale or even, gaining customer loyalty; a process called the conversion funnel (Figure 2). This runs parallel to the inbound marketing methodological model, which has four distinct phases: attract, convert, close and retain or seduce.

### 2.4. Attraction Phase

The attraction phase has a very important role in inbound marketing methodology. To attract a specific target audience, you must know what it is like in order to produce the appropriate content for all profiles, reaching users with information that has value because they are interested in it. To do this, inbound marketing proposes the structuring of the buyer persona which is the archetype of a service or product's ideal client that contains specific demographic data and information on aspects such as online, personal, professional and social behaviour and the relationship with the company that offers this product or service [21]. In short, it empathizes with potential users to find out what will tip their purchasing decisions.

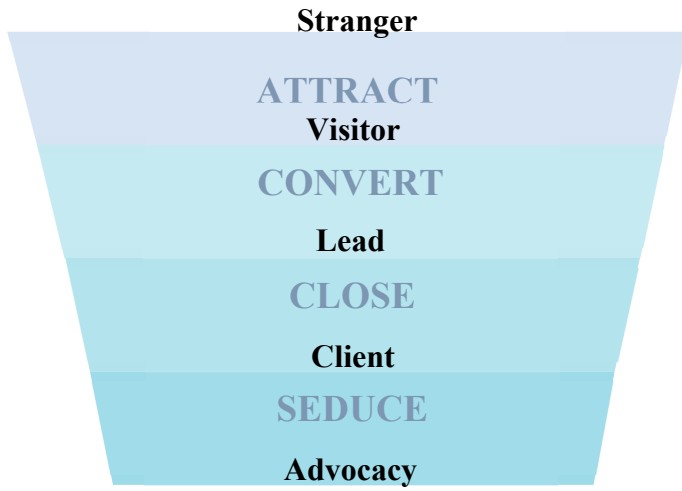

**Figure 2.** Conversion funnel. Source: García, 2014.

This phase is especially important in the current global tourism crisis brought on by the COVID-19 pandemic [30]. It is essential to attract the tourist again, to generate confidence when choosing the destination and for that role the key lies with inbound marketing.

Once the target audience is known, the appropriate resources and methods of transmission are determined, which are adapted from ad hoc content. There will be multiple contents, since visitors have different objectives: some only want information while others lead to conversion (from visitor to lead or register and from the latter to the client). Thus, the content strategy will be defined according to the different levels of the funnel [31], that is: A) TOFU (Top of the Funnel) or content focused on the attraction of traffic; B) MOFU (Middle of the Funnel) or content focused on giving reasons in order to find the solution users seek; C) BOFU (Bottom of the Funnel) or content focused on leading users to make a decision or take action (Figure 3).

Regarding sought-after traffic, the most valuable comes from primary data searches. i.e. searches for keywords, meaning that visits are of higher quality and, therefore, provide more visits. Moreover, direct traffic (users type the URL directly), social networks, references or e-marketing must be considered [32,33].

All the information extracted from the three levels of the funnel—TOFU, MOFU, BOFU—and the results of the sought-after and direct traffic could provide valuable data to generate the necessary confidence in the tourist necessary for overcoming the crisis being suffered worldwide, especially in Costa del Sol.

*2.5. Conversion Phase*

The conversion phase is part of the process that turns web traffic (the visitor) into leads on companies' databases, as they have vital contact information to start a commercial relationship. In this phase, the main tool is web analytics, which controls the results of different content in terms of conversion ratios (KPIs), so adjustments can be made in order to achieve maximum conversion ratios. Web analytics play a key role in the COVID crisis to convert virus-specific content into KPIs indicative of confidence- and security-building measures to bring tourists back to their traditional destination [34].

In addition, an education stage of leads begins that springs from marketing automation. Fundamentally, two techniques are used: A) Lead qualification: which assesses the leads' qualification in order to evaluate the probability of closing the purchasing cycle; B) Lead nurturing: personalization of the promotional content, according to each user's profile and where they are in the purchasing cycle phase.

Of the two techniques used in the conversion phase, we believe that the second one, lead nurturing, is more effective in the case of post-COVID tourist attraction, since it can directly affect the purchase

process via customizing the promotional content by adding elements that generate trust and security in the destination.

### 2.6. Closing and Loyalty Phase

The closing phase is where the sale actually takes place, although this is not the sole purpose of inbound marketing, since the seduction or loyalty stage is a continuation of the process, where customers who are kept satisfied and who continue to receive content they consider useful have to keep informed leads that, for different reasons, have not become clients, but that might be advocates of the brand on social networks.

The inbound marketing methodology requires careful synchronization so that all actions take place at the right time, and provide customers the content they need at that moment.

To have an inbound marketing strategy, we have already pointed out that there must be a total synchronicity between all actions undertaken, the user's qualification and at which stage of the purchasing process they are located.

It is necessary to define the life cycle of the contacts in the conversion funnel. In each of these phases, the information requested must be consolidated to evaluate the lead and choose the content to be sent to users (Figure 3).

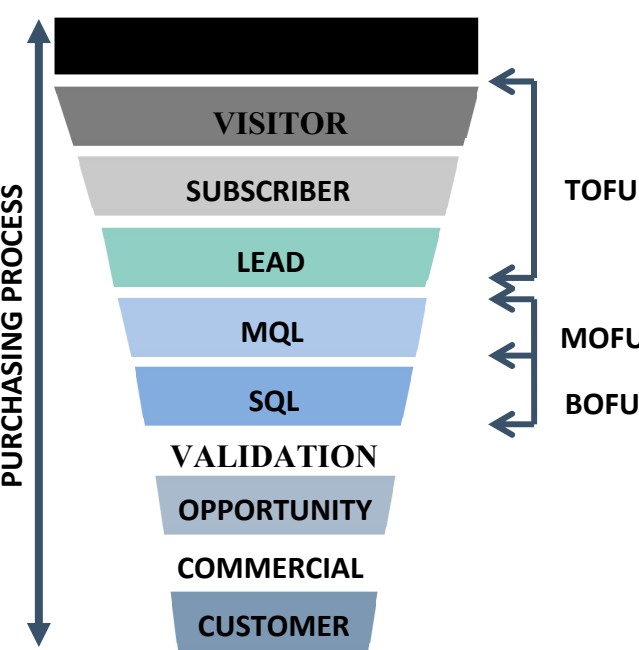

**Figure 3.** Life cycle of the contacts in the conversion funnel. Source: Costa del Sol Planning & Tourism Board, 2018 with supplements from the author.

For a visitor contact to be considered:

- a subscriber, an email address must have been given; with the current pandemic situation; this data is sufficient to send destination information to the subscriber.
- a lead (a first and last name) must have been given which will allow communications to be personalized. It is enough to have the first and second name to send personalized information about the COVID at the destination.
- a MQL (Marketing Qualified Lead), their interests (golf, culture, gastronomy, etc.) and their native country must be known. With this information, communications can be much more tailored. Personalized information regarding the COVID can be sent to these qualified contacts.

- a SQL (Sales Qualified Lead), besides the previous requirements, is fundamental to have detected the key factors in the user's trip arrangements; for example, the date of arrival at the destination. More restricted information about COVID can be made available to these SQL contacts.
- Finally, we reach the customer or client category. With specific customers you can send all the necessary information for your trip regarding COVID at the destination.

Parallel to the progress of the user in the conversion funnel, depending on their position in the funnel, the most appropriate content will be sent to each user according to the stage of the purchasing process they are in (Figure 3). With the crisis of COVID, more importance reaches the conversion funnel since content related to safety and confidence in the destination must be sent to the user.

At the TOFU level, content to attract traffic will be sent, aiming to increase the visibility of the Costa del Sol and convert the highest possible percentage of these people to subscribers and leads in order to feed the database. Actions are taken according to the blog content which is supplemented with new specific content on measures carried out to make the destination a safe and coronavirus-free place.

At the MOFU level, the content aims to provide information to answer users' questions or erase doubts and start the decision-making process. It will be more advanced content. The evergreen concept (content that will never be obsolete), now caters for all profiles, areas of interest and languages. Content related to safe and COVID-19-free sites should be integrated at this stage.

At the BOFU level, in situ content serves to build the loyalty of the users of the Costa del Sol strategy, and is aimed at those customers who are in the area and to whom the loyalty strategy can be applied. It is important to obtain information from those clients that after the crisis have decided to visit the Costa del Sol and have been able to verify in-situ the security of the destination free of COVID-19. They will act in their countries of origin as the best method of attracting new tourists, by applying the direct marketing tool "word of mouth".

Clearly, the content strategy is a fundamental piece with which to evaluate leads, and which can appear on different devices, in different formats and transmit different messages depending on the user's profile and position, now peppered with messages like: "Costa del Sol COVID free destination".

*2.7. Current Issues on Tourism and COVID-19*

We are currently witnessing an avalanche of scientific work on COVID-19, related to e-commerce, higher education, digital banking, etc. [35–37]. However, we agree with [38] that the coronavirus pandemic is unique and relevant to research and also that previous research on crises and disasters shows similar patterns that may well explain current phenomena. Furthermore, these authors warn us that existing knowledge may be subject to a paradigm shift in tourism due to the coronavirus pandemic.

This high volume of published papers makes sense due to the strong impact that the pandemic has had on the tourism sector. Hence, the rapid spatial diffusion of the COVID-19 epidemic outbreak has been revealed by [39] who have highlighted the total economic disruption of the Tourism Supply Chain, causing a significant reduction in revenue and creating liquidity issues for all operators. Various investigations have linked contagion and tourism [40], concluding that higher tourism is associated with increased mortality. This is a matter of debate as the pandemic continues. Investigations will follow one another and there will be a before and after in the COVID-19 crisis that will impact the tourism sector. Inbound marketing as digital methodology will help to revitalize tourist destinations.

## 3. Materials and Methods

*3.1. Case Study*

The case study method is a valuable research tool, and its greatest strength is that, through it, the behaviour of the people involved in the phenomenon studied is measured and recorded [41]. In this article we have adopted a contemporary case study methodology of a descriptive, holistic nature, which aims to highlight the factors that influence inbound marketing strategy. Based on the results

derived from the application of the case study, we are going to extrapolate them to the situation we are currently experiencing due to the world crisis caused by the coronavirus.

We think this is the most appropriate method of investigation since we are focusing on a new application in the tourism sector, studying the implementation of this strategy in a real situation under new circumstances derived from the emergence of the pandemic.

The inbound marketing methodology has been used in diverse sectors of the economy, from banking or insurance to the health sector, and includes education, mass consumption and professional services. The tourism field, in which the arrival of information and communication technologies has had a significant influence and, above all, presented numerous challenges [42], is not alien to this methodology, even though it is not yet widely applied [43].

The main objective of this research is to analyse the inbound marketing methodology and show how the Costa del Sol Planning & Tourism Board company is working on his model of relationship with customers under the concept of 'seduce the tourist', making an exploratory analysis of how these actions can be implemented for a post-COVID scenario to revitalise the tourism industry that is suffering so much from the effects of the pandemic. Thus, its performance in terms of corporate social responsibility develops initiatives that actively and voluntarily contribute to economic, social and environmental improvement, with the aim of adding value to the company's performance and achieving competitive advantages to face the two most important limitations of this field: seasonality and sustainability.

As stated in its action plan for this year, Costa del Sol Planning and Tourism [44] does not want to lose sight of the need to work on a destination policy focused on economic, social, cultural and environmental sustainability, to achieve better customer satisfaction, as well as continue using market intelligence to achieve more precise knowledge of the destination in which we want to act, and the profile of the client we are focusing on [45].

At the present time the application of this technology-based methodology is a major challenge in overcoming the global crisis. Since many jobs related to the tourism sector are at stake, social responsibility is a matter of high impact.

## 3.2. Starting Situation

Tourism & Planning Costa del Sol is a public company belonging to the Malaga Provincial Council whose objective is to promote the province as a tourist destination and support the growth of the economy through the design and execution of various projects. Besides being a technical instrument for the Council's committees, municipalities, associations and other organisms involved in the development of Malaga province.

The study of the current situation was undertaken using data on the variables of the Costa del Sol's ecosystem: generation of traffic, generation of leads and performance or behaviour of the company's database, taken from primary sources. These data were obtained by the inbound marketing project between April 2017 and March 2019—a period in which the process of the implementation strategy began—with traffic or leads from primary sources also considered [44].

A total of 4 meetings were held between the end of 2018 and the first quarter of 2019. The first of these was with the company's General Manager. The Director explained the digital strategy followed by Costa del Sol Planning and Tourism. This information was used to set the framework of the practical case we were going to make. It lasted approximately one hour. The other 3 meetings were with the company's Director of Digital Transformation and Tourism Promotion as the person responsible for digital management. These meetings were for field work so they were more extensive (about two hours each).

The destination competitiveness of Costa del Sol is closely related to the digital strategy Costa del Sol Planning & Tourism has adopted. The capacity for innovation and digital transformation is behind the qualitative and quantitative growth of the destination. At the last Agora Next International Conference, the public organization won an award for its leadership in destination intelligence.

## 4. Results and Discussion

The volume of traffic generated by the Costa del Sol ecosystem is sufficient to meet the main goals outlined in the strategy design: have one's public, convert the maximum number of users to leads that the target audience behaves positively towards the purchase.

### 4.1. Traffic Generation

Regarding traffic generation, the capacity of the Costa del Sol ecosystem to generate periodic and sustained traffic over time is extremely high, although it experienced a 15% decrease in traffic if we compare the first quarter of 2018 with the same period in 2019.

In addition, the peaks in traffic at third quarter of 2017, coincide with periods of heavy investment in advertising on the internet. In 2018, however, investment in internet advertising was lower, hence the difference between both quarters. Another important consideration is that the traffic operations are subject to a certain seasonality (in the second and third quarters, there is more traffic than in the rest of the quarters). Search engines are the main source of data traffic in the ecosystem and there is relatively low importance given to e-mail marketing (Table 2).

**Table 2.** Generation of total ecosystem traffic.

|  | 2Q 2017 | 3Q 2017 | 4Q 2017 | 1Q 2018 | 2Q 2018 | 3Q 2018 | 4Q 2018 | 1Q 2019 |
|---|---|---|---|---|---|---|---|---|
| Primary Data Search | 340.720 | 447.212 | 191.232 | 252.979 | 268.344 | 273.632 | 162.591 | 216.563 |
| Direct Traffic | 83.899 | 120.498 | 41.027 | 56.452 | 48.586 | 55.959 | 32.751 | 38.536 |
| Social Networks | 3.455 | 15.185 | 6.301 | 2.673 | 3.171 | 4.018 | 3.192 | 3.369 |
| References | 10.888 | 17.345 | 10.284 | 13.667 | 9.376 | 12.418 | 18.817 | 8.662 |
| E-Mail Marketing | 138 | 564 | 618 | 734 | 194 | 620 | 164 | 828 |
| Total | 439.100 | 600.804 | 249.462 | 326.505 | 329.671 | 346.647 | 217.515 | 267.958 |

Source: Planificación y Turismo Costa del Sol, 2019.

In line with the importance that content generation has as an inbound marketing strategy to attract a future customer or build loyalty once the purchase has been made, we can see that the Costa del Sol blog is a more dynamic channel, since there was an increase of 55% in traffic from search engines in the first quarter of 2019, compared to the same period in 2018, although the effects of the aforementioned seasonality were also felt, with the third quarter registering most visits (Table 3).

**Table 3.** Generation of blog traffic.

|  | 2Q 2017 | 3Q 2017 | 4Q 2017 | 1Q 2018 | 2Q 2018 | 3Q 2018 | 4Q 2018 | 1Q 2019 |
|---|---|---|---|---|---|---|---|---|
| Primary Data Search | 2.578 | 25.442 | 13.896 | 17.504 | 23.968 | 34.460 | 23.637 | 27.078 |
| Directo Traffic | 994 | 16.183 | 5.002 | 14.785 | 3.745 | 4.548 | 3.336 | 3.072 |
| Social Networks | 748 | 12.516 | 4.309 | 1.416 | 1.887 | 3.321 | 2.411 | 2.503 |
| References | 1.249 | 5.413 | 1.674 | 6.056 | 2.587 | 1.958 | 1.116 | 873 |
| E-Mail Marketing | 57 | 308 | 312 | 274 | 31 | 273 | 5 | 520 |
| Total | 5.626 | 59.862 | 25.193 | 40.035 | 32.218 | 44.560 | 30.505 | 34.046 |

Source: Planificación y Turismo Costa del Sol, 2019.

### 4.2. Contact Generation

Regarding contact generation, one of the challenges for inbound marketing is the conversion of anonymous users into leads, which involves them becoming part of the database and the necessary step to enable a direct relationship with future clients.

Discounting specific peaks, the Costa del Sol ecosystem generates between 400 and 600 quarterly contacts from its primary data sources (Table 4).

**Table 4.** Generation of total ecosystem contacts.

|  | **2Q 2017** | **3Q 2017** | **4Q 2017** | **1Q 2018** | **2Q 2018** | **3Q 2018** | **4Q 2018** | **1Q 2019** |
|---|---|---|---|---|---|---|---|---|
| Primary Data Search | 3 | 144 | 216 | 324 | 356 | 3450 | 300 | 413 |
| Direct Traffic | 15 | 194 | 79 | 85 | 380 | 85 | 80 | 78 |
| Social Networks | 3 | 90 | 33 | 11 | 8 | 6 | 9 | 29 |
| References | 7 | 33 | 116 | 49 | 115 | 46 | 63 | 43 |
| E-Mail Marketing | 1 | 1 | 0 | 3 | 9 | 14 | 13 | 1 |
| Total | 29 | 462 | 444 | 472 | 868 | 501 | 465 | 564 |

Source: Planificación y Turismo Costa del Sol, 2019.

According to the conversion data, in the last quarter of 2018 and the first quarter of 2019, the conversion ratio from visit to lead is 0.21% in primary sources, data that improves month by month and in the second quarter of 2018 reaches an important peak. The user consumes content, provides data during the decision-making stage of the trip and produces traffic at the same time. The third quarter of the year is the most dynamic in terms of the company's traffic and contact generation is more intense in the second quarter of the year, where there are key dates for planning holidays and when useful proposals are sought in order to choose a destination. After this time the user moves away from the ecosystem (Tables 4 and 5).

The validity of contacts is short (from 4 to 12 weeks) and is the maximum time available to achieve conversion.

**Table 5.** Conversion rate (%).

|  | **2Q 2017** | **3Q 2017** | **4Q 2017** | **1Q 2018** | **2Q 2018** | **3Q 2018** | **4Q 2018** | **1Q 2019** |
|---|---|---|---|---|---|---|---|---|
| Primary Data Search | 0.00% | 0.03% | 0.11% | 0.13% | 0.13% | 0.13% | 0.18% | 0.19% |
| Direct Traffic | 0.02% | 0.16% | 0.19% | 0.15% | 0.78% | 0.15% | 0.24% | 0.20% |
| Social Networks | 0.09% | 0.59% | 0.52% | 0.41% | 0.25% | 0.15% | 0.28% | 0.86% |
| References | 0.06% | 0.19% | 1.13% | 0.36% | 1.23% | 0.37% | 0.33% | 0.50% |
| E-Mail Marketing | 0.72% | 0.18% | 0.00% | 0.41% | 4.64% | 2.26% | 7.93% | 0.12% |
| Conversion Rate | 0.01% | 0.08% | 0.18% | 0.14% | 0.26% | 0.14% | 0.21% | 0.21% |

Source: Planificación y Turismo Costa del Sol, 2019.

*4.3. Database*

If we focus on the behaviour of the database, in the period of primary data collection (from the first quarter of 2017 to the first quarter of 2019), there were 11,447 records. Of these, only 9% have returned to enter the digital ecosystem, having captured 90% of these active users during the first quarter of 2018. In previous quarters, the most dynamic records have not exceeded 10% of the total. This data again reflects the fact that users use the web ecosystem when they are making their decisions and most do not return, since the ecosystem is unable to attract them and create the links for a more stable and lasting relationship. There is insufficient time to get the necessary information about users' interests and hobbies to be able to attract and retain them (Table 6).

**Table 6.** Percentage users returned to the ecosystem.

| Period | New Leads | Last Year Visits | Active (%) |
|--------|-----------|------------------|------------|
| 1Q 2017 | 2.428 | 38 | 1.6% |
| 2Q 2017 | 3.553 | 28 | 0.8% |
| 3Q 2017 | 485 | 46 | 9.5% |
| 4Q 2017 | 1.351 | 97 | 7.2% |
| 1Q 2018 | 685 | 98 | 14.3% |
| 2Q 2018 | 1.211 | 79 | 6.5% |
| 3Q 2018 | 637 | 58 | 9.1% |
| 4Q 2018 | 469 | 95 | 20.3% |
| 1Q 2019 | 628 | 571 | 90.9% |
| Total | 11.447 | 1.072 | 9.36% |

Source: Planificación y Turismo Costa del Sol, 2019.

Regarding the segmentation of users in relation to their position concerning the decision-making process (customer journey), the Costa del Sol database was graded low in relation to the life cycle of the contacts. 83% of these contacts are of the lowest category (subscribers) and only 14% are leads. There are no contacts categorised as MQL and SQL users. Trying to extrapolate these results to the current moment of the pandemic, we concluded that it is not possible to find continuity in the case study data; therefore, the current situation is totally unprecedented, and attracting tourists anew is critical. A novel digital ecosystem must be created for a different decision-making process post-COVID.

This means that those users which are actively making decisions about their holiday destination, or how many of them are currently at their destination, is unknown. Moreover, there is insufficient data on users for them to get personalized information based on their interests, so segmentation is impossible with this information, especially if applied to the current health crisis situation.

However, if we compare the data for 2015, in which the inbound marketing strategy was not implemented, with the year 2017 in which such a strategy existed, the indicators have improved significantly: the conversion rate in 2015 went from 14.5% to 31.8% in 2016. The average time users spent on a website was almost two minutes in 2015, increasing to more than six minutes in 2017, and the bounce rate fell by 59% in 2017, telephone contacts increased from 8,100 in 2015 to 23,000 in 2017 and tourist guides added to the itinerary increased from 134,000 in 2015 to 581,000 in 2017 (Table 7).

**Table 7.** Percentage users returned to the ecosystem.

| Indicators | 2015 | 2017 |
|------------|------|------|
| Conversion rate | 14.5% | 31.8% |
| Average time users spent on a website | 2 min. | 6 min. |
| Bounce rate | | Decreased 59% |
| Telephone contacts | 8100 | 23,000 |
| Tourist guides added to the itinerary | 134,000 | 581,000 |

Source: Planificación y Turismo Costa del Sol, 2019.

Extrapolating these results to the current situation of world crisis we conclude that the sustainable strategy of inbound marketing can facilitate attracting the tourist, seducing him and giving him security in the destination. Sending information regarding the situation that is being experienced at that moment in the destination can help to generate confidence in the traveller; hence, allow the stable relationship of mutual trust gained over time to be re-established.

## 5. Conclusions

The world is currently suffering the effects of a global economic crisis due to the incidence of the pandemic caused by COVID-19 [39]. At the tourism level, Spain, and especially the Costa del Sol, is being subjected to its devastating consequences. We believe that the sustainable inbound marketing

strategy, presented here as a case study, can be an effective solution in order to draw back tourists to those destinations visited before the Coronavirus crisis.

The evolution of the current market forces businesses to move from traditional or outbound marketing to inbound marketing, since it must adapt to the changes in customers' purchasing processes. However, in the tourism sector, companies implementing this methodology have not proliferated; hence, we resorted to a case study focusing on Costa del Sol Planning & Tourism Board as the first tourist destination in the world to implement a sustainable inbound marketing strategy.

The Costa del Sol Planning & Tourism Board focuses on the generation of new contacts, since leads lose their validity quickly (they move away from the system). Therefore, although inbound marketing advocates concentrating on the qualification of contacts so the database is permanently refined, and thus obtain only quality leads with which to interact, this is complicated in our example. It also makes it difficult to establish close relationships and supply relevant information adapted to the segment in question.

In fact, this shows that an ongoing inbound marketing strategy must respond rapidly in the 4-12-week decision-making period. There is room for improvement in the conversion rate of visit to lead, doubling the current conversion rates to around 1% in the short-medium term. Moreover, generating new contacts should be done more by way of increased conversion than by increased traffic. Work is being done to obtain better assets (blog entries, personalized information, etc.), to accelerate the evolution of contacts and qualify the greatest number of leads in those 12 weeks.

These latter results are not applicable in the current crisis but they may be indicative for the post-COVID period as deadlines may be shortened. This will be influenced by the quality of information provided to pre-crisis contacts or clients.

This fact is the key for the development of content and value propositions, from initial visibility to conversion. Advantage must be taken of users' interest in the decision phase to get to know them better and expand the database using qualitative criteria. We must bear in mind that sustainable inbound marketing actions are most appropriate when making medium- and long-term strategic decisions based on detailed knowledge of the potential client, especially considering the content furnished on COVID.

In addition, this continuous knowledge would open the destination further to more complex markets, such as the Nordic, Arab, Russian and Chinese, providing information adapted to the needs and idiosyncrasies of these potential customers, and drawing up specific action plans for each market [40]. In this sense, it is necessary to take into account the institutional policies of each country regarding the management of the coronavirus crisis, a determining factor in facilitating tourist movements between countries.

Another aim is to improve the quality of the content offered, which includes providing the automation typical of inbound marketing as we have suggested, but above all offering highly visual content and always converting into chunks for mobile phone viewing. All this is done from a much more detailed definition of the customer, in order to focus the marketing action on the target person interested in the Costa del Sol destination, adding the updated COVID-19 information.

The tourist continues to be at the center of any action, coordinating the strategy of inbound marketing with other projects launched by Costa del Sol Planning & Tourism Board, such as artificial intelligence, to detect business opportunities, or active listening in order to analyze what is happening in social networks, and, in general, with big data from the Destination Management System. When combining this data, the organization will get a much broader picture of the potential client, particularly in the case of customer acquisition and loyalty after the coronavirus tourism crisis.

The project indicators of inbound marketing that began in the second half of 2017 focused on contacts generated, qualified contacts, qualification rate, contacts converted into customers and loyal contacts. The results of the case study indicate that the current situation should be considered something new. It is impossible to find a trend or continuity with the data in this case; however, it can provide us with the idea that the post-COVID tourist must be treated as a new traveller who requires reliable,

safe, and constant information. It is therefore necessary to use the sustainable inbound marketing strategy to regain the customer, giving him greater confidence when compared to the situation before the pandemic.

Our analysis leaves us with important lessons from Spain's tourism boom before the COVID-19 crisis for the post-COVID-19 era phase of tourism revival. The most important conclusions are the following.

The rapid development of tourism in Spain before COVID-19 was largely due to the attraction of an increasingly segmented tourist industry that focused on the interests and tastes of the visitor. The result of the implementation of the Inbound Marketing strategy developed by Costa del Sol Planning and Tourism certifies this. In the phase of reactivation of tourism after COVID-19, a greater orientation towards tourist experiences could be useful, especially towards the unique experiences offered by Spain in which climate, culture and gastronomy are combined as success factors, differentiating associated products ("Spain brand"). To make this a reality, the information provided to the future tourist by inbound marketing strategy is decisive, especially if we take into account the probable changes in demand following COVID-19: regional (non-urban) tourism could well be more attractive to tourists than urban areas in the presence of COVID-19 and other health problems, giving priority to proximity to nature and low population density [46].

Furthermore, the use of an information policy on health-related issues, including the provision of health services in Spain as well as the health safety protocols could attract tourism.

Measures to restore travellers' confidence will play an important role in attracting tourists after crisis [47], to which the inbound marketing strategy provides a response, as it is based on contact with the future tourist through highly specialised content.

Future research could aim to determine whether this adaptation should be the same in all tourism subsectors, and how to deal with the short response time companies have to transform a web visitor into a loyal customer. COVID-19 also presents an opportunity for future research in tourism. A "before and after", or paradigm shift, will occur as a consequence of the appearance of the virus in the world, all of which will affect science at all levels and areas. This section is not mandatory, but can be added to the manuscript if the discussion is unusually long or complex.

The limitations of this research stem from the fact that there were no data on the tourism sector during the pandemic, making it impossible to compare the data provided by the inbound marketing strategy before the pandemic with those obtained during it.

**Author Contributions:** Conceptualization, E.M.S.-T. and M.R.-F.; methodology, E.M.S.-T.; writing—original draft preparation, E.M.S.-T., J.G.-M. and M.R.-F.; writing—review and editing, E.M.S.-T., J.G.-M. and M.R.-F. All authors have read and agreed to the published version of the manuscript.

**Funding:** This research received no external funding.

**Acknowledgments:** We would like to thank the company Costa del Sol Planning and Tourism for their help in preparing this paper.

**Conflicts of Interest:** The authors declare no conflict of interest.

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
