# Peer review of "The Application of the Inbound Marketing Strategy on Costa del Sol Planning & Tourism Board. Lessons for Post-COVID-19 Revival"

_sustainability, doi:10.3390/su12239926_

Round 1

Reviewer 1 Report

I believe that you should refer to a later version of the work of Robert Yin, you're actually referring to a version of his research methods handbook that dates back 30 years ago. As known, Yin has changed his mind about various steps of the design.

Author Response

Dear Reviewer1,

We will now answer all the questions raised by Reviewer1. We are grateful for your contributions, which have undoubtedly helped to improve the text considerably and we hope that you will be pleased with the new version.

Reviewer 1

I believe that you should refer to a later version of the work of Robert Yin, you're actually referring to a version of his research methods handbook that dates back 30 years ago. As known, Yin has changed his mind about various steps of the design.

We have consulted the latest version of Robert Yin's well-known book, which is the one we refer to in the text and which includes in the sixth edition of Robert K. Yin's best-selling text, a comprehensive manual on case study research. We have changed the citation and the reference for the latest version used.

Reviewer 2 Report

I really enjoyed reading your article. I only suggest that you develop the chapter of the conclusions, namely discussing the results, indicating the implications for science, organizations/companies, and pointing out the limitations and suggestions for future research.

Author Response

Dear Reviewer 2

Thanks a lot for the time and the effort in revising our paper. We have performed the changes according to your suggestions.

Your comments

I really enjoyed reading your article. I only suggest that you develop the chapter of the conclusions, namely discussing the results, indicating the implications for science, organizations/companies, and pointing out the limitations and suggestions for future research.

Response

Thank you very much for your comment. We have included more ideas in the sections results and conclusions; the implications for the society, in general, have been now added. Limitations of the study and future research also appears in the final section of the article.

We hope the paper now comply with all your requirements.

With our best wishes,

The authors

Reviewer 3 Report

  1. Originality/Novelty:

This analysis is useful because it provides an overview and understanding of the type and scale of the COVID-19 tourism impacts.

  1. Significance: 

This study describe practical and theoretical implications on how to better research, understand the Covid crises in tourism industry. Data sources and forecasts have shifted, and proliferated, in the crisis. The tourism sector needs to undergo an analytics transformation to enable the coordination of marketing budgets, calendars of events, and to ensure that products are marketed to the right population segment at the right time in complete safety condition.

  1. Are the results interpreted appropriately? Are they significant? Are all conclusions justified and supported by the results? Are hypotheses and speculations carefully identified as such?

The present analysis is not exhaustive in terms of the COVID-19 impacts. For example, the COVID-19 has different impacts on tourism operators based on their characteristics such as, the nature of the tourism sector (intermediaries, event organizers transportation, type of accommodation or attraction provider), their size, location, management and ownership style.

  1. Quality of Presentation: Is the article written in an appropriate way?

In this paper, 59-66 lines are copy/paste after Instruction of Authors, I think it is a mistake!!!

Inbound marketing vs outbound marketing (2.2.) should be summarized in a table and related with realities in Covid period.

  1. Are the data and analyses presented appropriately? Are the highest standards for presentation of the results used?

Research investigating, measuring and predicting the COVID-19 tourism impacts should be improved.

In the article are analyse only the dates pre-crises. The relevance of these data (2017-2019)  in the context of the Covid crisis should be better justified, I am not sure that the research is relevant for the situation of tourism during the Covid period.

The analysis of circumstances of COVID-19 (e.g. stay at home lockdowns, social distancing) missed

  1. Scientific Soundness: 

I doubt if the data is robust enough to draw the conclusions.

I don’t consider  the research investigating are  described with sufficient details to allow another researcher to reproduce the results. A framework is not an absolute necessity, but nonetheless it is very useful.

  1. Interest to the Readers:

The paper could attract a wide readership, if take into consideration the persistent digital divide found in consumers and tourism businesses  and the analysis which converted the pandemic to an infodemic (e.g. lack or mis-information, diffusion of fake COVID-19 news and advises, emotional contagion of global depression and mental health)

The analysis did not also include other major tourism stakeholders such as tourism employees, local communities, tourism entrepreneurs and tourism education (scholars, students and institutions alike).

  1. English Level: Is the English language appropriate and understandable?

The English level is appropriate, but the English language should be improved

Author Response

Dear Reviewer3,

We will now answer all the questions raised by Reviewer3. We are grateful for your contributions, which have undoubtedly helped to improve the text considerably and we hope that you will be pleased with the new version.

Reviewer 3

Comments and Suggestions for Authors

Originality/Novelty:

This analysis is useful because it provides an overview and understanding of the type and scale of the COVID-19 tourism impacts.

Significance:

This study describe practical and theoretical implications on how to better research, understand the Covid crises in tourism industry. Data sources and forecasts have shifted, and proliferated, in the crisis. The tourism sector needs to undergo an analytics transformation to enable the coordination of marketing budgets, calendars of events, and to ensure that products are marketed to the right population segment at the right time in complete safety condition.

Are the results interpreted appropriately? Are they significant? Are all conclusions justified and supported by the results? Are hypotheses and speculations carefully identified as such?

The present analysis is not exhaustive in terms of the COVID-19 impacts. For example, the COVID-19 has different impacts on tourism operators based on their characteristics such as, the nature of the tourism sector (intermediaries, event organizers transportation, type of accommodation or attraction provider), their size, location, management and ownership style.

Quality of Presentation: Is the article written in an appropriate way?

In this paper, 59-66 lines are copy/paste after Instruction of Authors, I think it is a mistake!!!

We have eliminated this paragragh because it was a mistake.

Inbound marketing vs outbound marketing (2.2.) should be summarized in a table and related with realities in Covid period.

We have created the table and added in the text (section 2.2).

Are the data and analyses presented appropriately? Are the highest standards for presentation of the results used?

Research investigating, measuring and predicting the COVID-19 tourism impacts should be improved.

In the article are analyse only the dates pre-crises. The relevance of these data (2017-2019)  in the context of the Covid crisis should be better justified, I am not sure that the research is relevant for the situation of tourism during the Covid period.

The analysis of circumstances of COVID-19 (e.g. stay at home lockdowns, social distancing) missed

Scientific Soundness:

I doubt if the data is robust enough to draw the conclusions.

We explain in the text the link between inbound marketing data from before the COVID-19 period and how this strategy could be used for the post-COVID-19 period.

I don’t consider  the research investigating are  described with sufficient details to allow another researcher to reproduce the results. A framework is not an absolute necessity, but nonetheless it is very useful.

This article is a case study not a research paper. It explains the methodology followed in a case study and the results obtained.

Interest to the Readers:

The paper could attract a wide readership, if take into consideration the persistent digital divide found in consumers and tourism businesses  and the analysis which converted the pandemic to an infodemic (e.g. lack or mis-information, diffusion of fake COVID-19 news and advises, emotional contagion of global depression and mental health)

Thanks you for your comments

The analysis did not also include other major tourism stakeholders such as tourism employees, local communities, tourism entrepreneurs and tourism education (scholars, students and institutions alike).

We have introduced the impacts that COVID 19 has on the different stakeholders in the tourism sector.

English Level: Is the English language appropriate and understandable?

The English level is appropriate, but the English language should be improved

A new proofreading has been carried out.

Reviewer 4 Report

In my opinion the paper has no value to be published in this journal. I suggest to rethink to the structure and aim of the paper and then try to submit again, following all the comments provided. Here are in detail my motivations:

  • The abstract misses the results of the research and the conclusions of the study
  • How does it mean this part at page 2 from line 59 to 66? “The introduction should briefly place the study in a broad context and highlight why it is important. It should define the purpose of the work and its significance. The current state of the research field should be reviewed carefully and key publications cited. Please highlight controversial and diverging hypotheses when necessary. Finally, briefly mention the main aim of the work and highlight the principal conclusions. As far as possible, please keep the introduction comprehensible to scientists outside your particular field of research. References should be numbered in order of appearance and indicated by a numeral or numerals in square brackets, e.g., [1] or [2,3], or [4–6]. See the end of the document for further details on references.”
  • The introduction section is inconsistent: the aim of the paper is not stated, nor the gap that the study wants to cover and the added value. What is more, it is strongly suggested that Authors develop research questions for the paper, to better understand what they are searching for.
  • The way references are cited is not correct, please follow journal guidelines for this.
  • The literature review paragraph is all focused on inbound marketing: its definition, the difference with outbound marketing and all the process. I believe that this part is too academic, and adds nothing more to the work. The functioning of inbound marketing is already well known and does not need to occupy all this space. I believe it is important that it will be briefly summarized in a single paragraph and then leave space instead for the analysis of the tourism sector, the impact of covid-19 on this sector and how new strategies including inbound marketing can be of absolute interest for the sector. It can be created a section where it is described in which sectors the inbound marketing has been already adopted, including the tourism sector. Authors need to focus the literature review section on these very recent and current issues to make the work of interest.
  • In the methodology section nothing has been said about how the data for the case study have been obtained. It is extremely important to describe in detail how data have been gathered and how Authors have analyzed them.
  • In the result section Authors do not permit to understand clearly the link between these data on inbound marketing that are pre covid-19 period and how this strategy could be used in this period. Results are totally inconsistent with the title of the study. Why an inbound strategy could be an effective solution? No motivations are given. Then Authors say that it could be important for post-covid period, but the results do not permit to reach this conclusion. this is the authors' thinking, but it is not supported by results. In my opinion this is nor a research paper, it is an opinion paper, therefore I cannot accept it for publication in this important journal.

Author Response

Dear Reviewer4,

We will now answer all the questions raised by Reviewer4. We are grateful for your contributions, which have undoubtedly helped to improve the text considerably and we hope that you will be pleased with the new version.

Reviewer 4

In my opinion the paper has no value to be published in this journal. I suggest to rethink to the structure and aim of the paper and then try to submit again, following all the comments provided. Here are in detail my motivations:

  • The abstract misses the results of the research and the conclusions of the study
  • How does it mean this part at page 2 from line 59 to 66? “The introduction should briefly place the study in a broad context and highlight why it is important. It should define the purpose of the work and its significance. The current state of the research field should be reviewed carefully and key publications cited. Please highlight controversial and diverging hypotheses when necessary. Finally, briefly mention the main aim of the work and highlight the principal conclusions. As far as possible, please keep the introduction comprehensible to scientists outside your particular field of research. References should be numbered in order of appearance and indicated by a numeral or numerals in square brackets, e.g., [1] or [2,3], or [4–6]. See the end of the document for further details on references.”

We have eliminated this paragraph because it was a mistake.

  • The introduction section is inconsistent: the aim of the paper is not stated, nor the gap that the study wants to cover and the added value. What is more, it is strongly suggested that Authors develop research questions for the paper, to better understand what they are searching for.
  • The way references are cited is not correct, please follow journal guidelines for this.

An exhaustive review of references has been made following the guidelines of the journal.

  • The literature review paragraph is all focused on inbound marketing: its definition, the difference with outbound marketing and all the process. I believe that this part is too academic, and adds nothing more to the work. The functioning of inbound marketing is already well known and does not need to occupy all this space. I believe it is important that it will be briefly summarized in a single paragraph

It is very difficult to summarize this whole section in one paragraph. We want to emphasize that this is a case study, thus we have to explain what the methodology used consists of. It is not only a question of explaining what inbound marketing consists of, but also the relationship with the application of this methodology in the post-COVID-19 era.

and then leave space instead for the analysis of the tourism sector, the impact of covid-19 on this sector and how new strategies including inbound marketing can be of absolute interest for the sector. It can be created a section where it is described in which sectors the inbound marketing has been already adopted, including the tourism sector. Authors need to focus the literature review section on these very recent and current issues to make the work of interest.

However, we have included the section that requested us.

  • In the methodology section nothing has been said about how the data for the case study have been obtained. It is extremely important to describe in detail how data have been gathered and how Authors have analyzed them.

The narrative of this study presents the Spanish company Costa del Sol Planning and Tourism, which has successfully implemented inbound actions. All the data provided here have been supplied by this company in different meetings held. They reflect the results of the first phase of the implementation of the inbound marketing actions. 

  • In the result section Authors do not permit to understand clearly the link between these data on inbound marketing that are pre covid-19 period and how this strategy could be used in this period. Results are totally inconsistent with the title of the study. Why an inbound strategy could be an effective solution? No motivations are given. Then Authors say that it could be important for post-covid period, but the results do not permit to reach this conclusion. This is the authors' thinking, but it is not supported by results. In my opinion this is not a research paper, it is an opinion paper, therefore I cannot accept it for publication in this important journal.

This article is a case study not a research paper. It explains the methodology followed in a case study and the results obtained. In the text we justify, according to different institutions and authors, that the results obtained with a digital marketing strategy such as inbound marketing, and specifically that applied in the company Costa del Sol Planning and Tourism, can be a very useful tool in the revitalisation of the tourism sector once the current health crisis has passed.

Round 2

Reviewer 3 Report

 I'm pleased with the new version.

Author Response

Dear Reviewer 3,

We have re-read and corrected the errors in the English language that we have detected.

Thank you very much for the time and effort you have put into our manuscript. 

Sincerely, 

The authors

Reviewer 4 Report

The paper in my opinión needs other revisions before it is ready for publication. Now authors have better underlined that the data obtained have nothing to do with the pandemic period. This was very confusing and misleading in the previous versión of the work. In detail my comments:

  • Language should be revised, the text is full of grammatical errors.
  • Acronyms should be written in capital letters (eg. Table 1)
  • Figure 3 should be revised; some text is not well seen (eg. Validation, commercial…)
  • References are not cited in the correct way as told before (eg. Use square brackets). Please read with attention the guidelines for authors.
  • I made a previous comment on the methodology “In the methodology section nothing has been said about how the data for the case study have been obtained. It is extremely important to describe in detail how data have been gathered and how Authors have analyzed them.” And the Authors answered me like this “The narrative of this study presents the Spanish company Costa del Sol Planning and Tourism, which has successfully implemented inbound actions. All the data provided here have been supplied by this company in different meetings held. They reflect the results of the first phase of the implementation of the inbound marketing actions. “ It is really important to have a correct methodology section that Authors define how many meetings did they do with the company, in which period and how long did they last. Please insert in detail how data and information have been gathered.

Author Response

Dear Reviewer 4,

Above all, we would like to thank you for the time and effort you have put into correcting our work.
According to your instructions, we have made the following changes:

The English language has been revised.
Acronyms have been written in capital letters.
Figure 3 has been improved.
References have been corrected.
In the methodology section, it has been explained more extensively how the data for the case study have been obtained.
Let us know if we need to make any other corrections.

Best regards,

The authors